# CARMS: Categorical-Antithetic-REINFORCE Multi-Sample Gradient Estimator

**Alek Dimitriev**
McCombs School of Business
The University of Texas at Austin
Austin, TX 78712
alek.dimitriev@mccombs.utexas.edu

**Mingyuan Zhou**
McCombs School of Business
The University of Texas at Austin
Austin, TX 78712
mingyuan.zhou@mccombs.utexas.edu

## Abstract

Accurately backpropagating the gradient through categorical variables is a challenging task that arises in various domains, such as training discrete latent variable models. To this end, we propose CARMS, an unbiased estimator for categorical random variables based on multiple mutually negatively correlated (jointly antithetic) samples. CARMS combines REINFORCE with copula based sampling to avoid duplicate samples and reduce its variance, while keeping the estimator unbiased using importance sampling. It generalizes both the ARMS antithetic estimator for binary variables, which is CARMS for two categories, as well as LOORF/VarGrad, the leave-one-out REINFORCE estimator, which is CARMS with independent samples. We evaluate CARMS on several benchmark datasets on a generative modeling task, as well as a structured output prediction task, and find it to outperform competing methods including a strong self-control baseline. The code is publicly available.[1]

## 1 Introduction

When optimizing an expectation based objective of the form $\mathbb{E}_{z\sim q_\phi(z)}[f(z)]$, we sometimes require the gradients with respect to the parameters $\phi$ of the distribution. This is challenging for discrete variables, because the commonly used reparameterization gradient does not directly work, unless the discrete distribution is approximated by a continuous and reparameterizable one [Jang et al., 2017, Maddison et al., 2017]. A significant part of this field has thus been score function (REINFORCE) based estimators [Glynn, 1990, Williams, 1992, Fu, 2006], which are general and do not require $f$ to be the differentiable. In this paper, we focus on the case when $z$ is a high dimensional categorical variable with logits $\phi$. An common example of this form is the evidence lower bound (ELBO) [Jordan et al., 1998], which arises in variational inference and is used for training variational autoencoders [Kingma and Welling, 2014, Rezende et al., 2014]. Because their latent space consists of a large number of categorical variables, they require Monte Carlo gradients with respect to the parameters of the stochastic distribution, but have excellent performance, as a categorical VAE has in practice achieved state-of-the-art zero-shot image generation [Ramesh et al., 2021].

Our main contribution is a novel unbiased and low-variance gradient estimator for categorical variables. The Categorical-Antithetic-REINFORCE-Multi-Sample (CARMS) estimator uses a copula to generate any number of antithetic (mutually negatively correlated) [Owen, 2013] categorical samples, and constructs an unbiased estimator by combining them into a baseline for variance reduction. This approach is inspired by the ARMS estimator [Dimitriev and Zhou, 2021], which also uses multiple antithetic samples, but only works for binary variables. For two categories, CARMS

---

[1] https://github.com/alekdimi/carms

reduces to it, while for independent samples, CARMS reduces to the leave-one-out-REINFORCE (LOORF) estimator. Our approach achieves higher ELBO with VAE models than the state of the art, as well as higher log likelihood for conditional image completion.

**Related work**   One widely used group of gradient estimators for categorical variables is based on trading off bias for lower variance. This includes the straight through (ST) estimator [Bengio et al., 2013], the direct argmax [Lorberbom et al., 2019], as well as the concurrently developed equivalent Gumbel-Softmax (GS) [Jang et al., 2017] or Concrete [Maddison et al., 2017] that uses a continuous relaxation. These were further improved by combining them with REINFORCE to obtain unbiased estimators, which includes REBAR [Tucker et al., 2017], as well as RELAX [Grathwohl et al., 2018], which uses a free-form neural network.

In practice, REINFORCE is almost always augmented by baselines, *e.g.*, in variational inference [Mnih and Gregor, 2014, Ranganath et al., 2014, Paisley et al., 2012, Ruiz et al., 2016, Kucukelbir et al., 2017]. MuProp [Gu et al., 2016] uses a first order mean-field Taylor approximation, but requires $f$ to be differentiable. Other estimators apply Rao-Blackwellization [Liu et al., 2019, Kool et al., 2020], *e.g.*, to one latent variable at a time [Titsias and Lázaro-Gredilla, 2015], to the ST estimator [Paulus et al., 2021], or to the reparameterized Dirichlet vector in ARSM [Dong et al., 2021]. Besides REINFORCE and reparameterization type gradients, there is also the measure valued gradient [Rosca et al., 2019] and finite differences [Fu, 2006], but are less common in practice. A comprehensive review can be found in Mohamed et al. [2020].

The more recent approaches for categorical variables are unbiased and REINFORCE based, building on the idea of using multiple samples to construct a baseline for variance reduction. One such baseline is LOORF [Kool et al., 2019a], originally introduced in Salimans and Knowles [2014] and also known as VarGrad [Richter et al., 2020], where its theoretical properties are further analyzed. VIMCO [Mnih and Rezende, 2016] has a similar form, but is specific to the importance weighted multi sample bound [Burda et al., 2016]. A different approach is ARSM [Yin et al., 2019], which reparameterizes the gradient with a Dirichlet distribution and uses swaps to obtain multiple correlated samples. However, it adds some amount of variance due to the continuous reparameterization, and can require up to $C(C-1)/2$ function evaluations per step, which can be computationally expensive. More recently, the unordered set estimator (UNORD) [Kool et al., 2020] uses the Gumbel top-k trick to sample without replacement, and then constructs a baseline using a multiplicative term to preserve its unbiasedness.

## 2   Background

Let $\boldsymbol{z} = (\boldsymbol{z}_1, ..., \boldsymbol{z}_D)$ denote $D$ independent categorical variables, where $\boldsymbol{z}_d$ is a one hot encoded categorical sample from $\boldsymbol{z}_d \sim q_{\boldsymbol{\phi}_d}(\boldsymbol{z}_d) = \mathrm{Cat}(\sigma(\boldsymbol{\phi}_d))$, with $\sigma(\boldsymbol{x})_i = e^{\boldsymbol{x}_i} / \sum_j e^{\boldsymbol{x}_j}$ being the softmax function. Let also $\hat{\boldsymbol{\imath}}$ be the basis vector with its $i^{\text{th}}$ coordinate set to one, and otherwise zero, such that $P(\boldsymbol{z} = \hat{\boldsymbol{\imath}}) = \sigma(\boldsymbol{\phi})_i$, or equivalently $\mathbb{E}[\boldsymbol{z}] = \sigma(\boldsymbol{\phi})$. Unless otherwise stated, superscripts denote the dimension, and subscripts denote different samples, with vectors and matrices being bold lowercase and bold uppercase symbols, respectively. We are interested in optimizing the following objective with respect to the logits $\boldsymbol{\phi} = (\boldsymbol{\phi}_1, ..., \boldsymbol{\phi}_D)$:

$$\mathcal{L}(\boldsymbol{\phi}) = \mathbb{E}_{\boldsymbol{z} \sim q_{\boldsymbol{\phi}}(\boldsymbol{z})}[f(\boldsymbol{z})], \quad q_{\boldsymbol{\phi}}(\boldsymbol{z}) = \prod_{d=1}^{D} q_{\boldsymbol{\phi}_d}(\boldsymbol{z}_d).$$

Although the score function gradient contains two terms:

$$\nabla_{\boldsymbol{\phi}} \mathcal{L}(\boldsymbol{\phi}) = \mathbb{E}_{\boldsymbol{z} \sim q_{\boldsymbol{\phi}}(\boldsymbol{z})} \Big[ \nabla_{\boldsymbol{\phi}} f(\boldsymbol{z}) + f(\boldsymbol{z}) \nabla_{\boldsymbol{\phi}} \ln q_{\boldsymbol{\phi}}(\boldsymbol{z}) \Big]$$

the first term is easily estimated, so we omit the subscript in $f$ for notational clarity, and we focus on the latter term in this work. Since CARMS generalizes the multisample LOORF estimator beyond independent samples, we review LOORF here. We also review the ARMS estimator, which uses a copula to generate antithetic samples, but is restricted to binary variables.

## 2.1 LOORF

Leave-one-out-REINFORCE (LOORF) [Salimans and Knowles, 2014, Kool et al., 2019a], also known as VarGrad [Richter et al., 2020], is a general score function based gradient estimator that uses $N$ i.i.d. samples to construct a baseline for variance reduction, and is competitive with state of the art estimators. Its theoretical properties have recently been analyzed in Richter et al. [2020], where the same estimator results from minimizing the variance of the log ratio between the posterior and approximating distribution in variational inference. The only requirements for LOORF are the ability to sample $\boldsymbol{z} \sim q_\phi(\boldsymbol{z})$ and evaluate $f(\boldsymbol{z})$. Given $N$ samples $\boldsymbol{z}_1, ..., \boldsymbol{z}_N \overset{iid}{\sim} \mathrm{Cat}(\sigma(\boldsymbol{\phi}))$, it has the form:

$$g_{\mathrm{LOORF}} = \frac{1}{N-1} \sum_{n=1}^{N} \left( f(\boldsymbol{z}_n) - \bar{f}(\boldsymbol{z}) \right) \nabla_\phi \ln q_\phi(\boldsymbol{z}_n) = \frac{1}{N-1} \sum_{n=1}^{N} \left( f(\boldsymbol{z}_n) - \bar{f}(\boldsymbol{z}) \right) (\boldsymbol{z}_n - \sigma(\boldsymbol{\phi}_n)), \quad (1)$$

where $\bar{f}(\boldsymbol{z}) = \frac{1}{N} \sum_{n=1}^{N} f(\boldsymbol{z}_n)$. It is commonly used due to its simplicity and strong performance.

## 2.2 ARMS

The Antithetic-REINFORCE-MultiSample (ARMS) [Dimitriev and Zhou, 2021] estimator is a recent work that uses any number $N$ of mutually negatively correlated samples. However, although unbiased, it is limited to binary variables, which motivated us to extend it to the categorical case. Since any antithetic copula in two dimensions reduces to the pair $(u, 1 - u)$, it generalizes DisARM [Dong et al., 2020], independently discovered as unbiased uniform gradient (U2G) [Yin et al., 2020], which use $N = 2$ samples. ARMS achieves this generalization by using a copula [Trivedi and Zimmer, 2007], which is any multivariate distribution whose marginals are uniform random variables:

$$\boldsymbol{u} = (u_1, ..., u_N) \sim \mathcal{C}_N, \quad \forall i: \ u_i \sim \mathrm{Unif}(0, 1).$$

ARMS uses a Dirichlet or Gaussian copula with strong negative dependence between each dimension of $\boldsymbol{u}$. However any copula can be used instead, with the only two requirements being the ability to generate samples easily, as well as being able to evaluate the bivariate CDF $\Phi(\boldsymbol{u}_i, \boldsymbol{u}_j)$ of the copula, so that the debiasing term can be calculated. Given a copula sample $\boldsymbol{u}$, it uses inverse CDF sampling to convert it into $N$ antithetic $\mathrm{Bern}(p)$ samples, which are simply $b_i = \mathbb{1}_{u_i < p}, \ \forall i$. For a set of $N$ antithetic $D$-dimensional Bernoulli variables $\boldsymbol{b}_1, ..., \boldsymbol{b}_N$ with probabilities $\sigma(\boldsymbol{\phi})$, the $N$-sample unbiased estimator has the following simple form:

$$g_{\mathrm{ARMS}} = \frac{1}{N-1} \sum_{n=1}^{N} \left( f(\boldsymbol{b}_n) - \frac{1}{n} \sum_{m=1}^{N} f(\boldsymbol{b}_m) \right) \frac{\boldsymbol{b}_n - \sigma(\boldsymbol{\phi})}{1 - \boldsymbol{\rho}}, \quad (2)$$

with $\boldsymbol{\rho} = (\rho_1, ..., \rho_D)$ and $\rho_d = \mathrm{corr}(\boldsymbol{b}_i^d, \boldsymbol{b}_j^d)$, which is simple to compute given the bivariate CDF of the copula.

## 3 CARMS

The CARMS estimator has a similar form to LOORF, but requires a multiplicative term to remain unbiased. It has two requirements: an easy way to sample antithetic categorical variables, and being able to compute the bivariate probability mass function (PMF), which should be identical for any pair. For clarity, we begin by assuming the ability to do this, and derive the univariate version for two samples, which we then extend to $N$ samples. Next, we generalize CARMS to any number of categorical variables. Lastly, in Section 3.2, we show two different ways of sampling that satisfy both conditions: inverse CDF sampling with an easily computable analytical bivariate PMF, and the Gumbel max trick combined with an empirical estimate of the PMF.

The two sample version of CARMS can be obtained by replacing the two $i.i.d.$ samples in 2-LOORF with an arbitrarily correlated pair of categorical variables and an added debiasing term. For $N = 2$ samples $z, z' \sim \mathrm{Cat}(\sigma(\boldsymbol{\phi}))$, LOORF has the following simple form:

$$g_{\mathrm{LOORF}}(\boldsymbol{z}, \boldsymbol{z}') = \frac{1}{2} \Big( f(\boldsymbol{z}) - f(\boldsymbol{z}') \Big)(\boldsymbol{z} - \boldsymbol{z}'), \quad (3)$$

which, importantly, is unbiased [Kool et al., 2019a, Dimitriev and Zhou, 2021]. This means that using a simple importance weight preserves its unbiasedness, for an arbitrary bivariate categorical distribution $(z, z') \sim \text{Cat}(\sigma(\phi), \sigma(\phi))$:

$$\mathbb{E}\left[g_{\text{LOORF}}(z, z')\frac{P(z = \hat{\imath})P(z' = \hat{\jmath})}{P(z = \hat{\imath}, z' = \hat{\jmath})}\right] = \sum_{i,j} P(z = \hat{\imath}, z' = \hat{\jmath})\frac{P(z = \hat{\imath})P(z' = \hat{\jmath})}{P(z = \hat{\imath}, z' = \hat{\jmath})}g_{\text{LOORF}}(\hat{\imath}, \hat{\jmath})$$

$$= \sum_{i,j} P(z = \hat{\imath})P(z' = \hat{\jmath})g_{\text{LOORF}}(\hat{\imath}, \hat{\jmath}) = \mathbb{E}_{z,z' \overset{iid}{\sim} \text{Cat}(\sigma(\phi))}\left[g_{\text{LOORF}}(z, z')\right] = \nabla_\phi \mathcal{L}(\phi).$$

We summarize the derivation of the two sample version of CARMS, which we denote as the Categorical-Antithetic-REINFORCE-Two-Sample (CARTS) estimator in the following theorem.

**Theorem 1** *Let $(z, z')$ be a sample from an arbitrary bivariate Categorical distribution with marginal distributions $z, z' \sim \text{Cat}(\sigma(\phi))$, and a known bivariate PMF. An unbiased estimator of $\nabla_\phi \mathbb{E}[f(z)]$ is:*

$$g_{\text{CARTS}}(z, z') = \frac{1}{2}\Big(f(z) - f(z')\Big)(z - z')z^T \mathcal{R} z', \quad \mathcal{R}_{ij} = \frac{\sigma(\phi)_i \sigma(\phi)_j}{P(z = \hat{\imath}, z' = \hat{\jmath})}. \tag{4}$$

The intuition behind using negatively correlated variables is that we want to avoid the case when $z = z'$, because the sample is then "wasted." We formalize this intuition below, and defer the proof to the Appendix.

**Theorem 2** *Let $(z, z') \sim \text{Cat}(\sigma(\phi), \sigma(\phi))$. If the bivariate PMF satisfies:*

$$\forall i \neq j : P(z = \hat{\imath}, z' = \hat{\jmath}) \geq P(z = \hat{\imath})P(z' = \hat{\jmath}),$$

*with strict inequality for at least one pair, then $\text{Var}[g_{CARTS}(z, z')] < \text{Var}[g_{LOORF}(z, z')]$.*

We now extend CARTS to $N$ samples, using the following identity, the proof of which can be found in Dimitriev and Zhou [2021]. It states that $N$-sample LOORF is equivalent to averaging 2-sample LOORF over all $\binom{N}{2}$ pairs:

$$g_{\text{LOORF}}(z_1, .., z_N) = \frac{1}{N}\sum_{n=1}^{N}\left(f(z_n) - \frac{1}{N}\sum_{m=1}^{N} f(z_m)\right)(z_n - \sigma(\phi))$$

$$= \frac{1}{N(N-1)}\sum_{n \neq m} \frac{1}{2}\Big(f(z_n) - f(z_m)\Big)(z_n - z_m) = \frac{1}{N(N-1)}\sum_{n \neq m} g_{\text{LOORF}}(z_n, z_m).$$

With the above identity it easily follows that given $N$ antithetic categorical samples $Z = [z_1, ..., z_N]^T$, applying CARTS to all pairs results in an unbiased estimator, due to linearity of expectations:

$$\mathbb{E}\left[g_{\text{CARMS}}(Z, \mathcal{R})\right] = \mathbb{E}\left[\frac{1}{N(N-1)}\sum_{n \neq m} g_{\text{CARTS}}(z_n, z_m, \mathcal{R})\right] = \mathbb{E}\left[\frac{1}{N(N-1)}\sum_{n \neq m} g_{\text{LOORF}}(z_n, z_m)\right]$$

$$= \mathbb{E}\left[g_{\text{LOORF}}(Z)\right] = \nabla_\phi \mathcal{L}(\phi).$$

We summarize CARMS in the next theorem and also rewrite it in a simpler matrix form used in our implementation. The matrix form can be obtained after some algebra, which can be found in the Appendix.

**Theorem 3** *Let $Z = [z_1, ..., z_N]^T$ be a sample from an arbitrary $N$-variate Categorical distribution with identical marginal and bivariate distributions, such that $z_i \sim \text{Cat}(\sigma(\phi))$, and $\mathcal{R}_{ij} = \sigma(\phi)_i \sigma(\phi)_j / P(z_n = \hat{\imath}, z_m = \hat{\jmath})$. An unbiased estimator of $\nabla_\phi \mathbb{E}[f(z)]$ is:*

$$g_{CARMS}(Z) = \frac{1}{N(N-1)}\sum_{n \neq m} \frac{1}{2}\Big(f(z_n) - f(z_m)\Big)(z_n - z_m)z_n^T \mathcal{R} z'_m$$

**Lemma 4** *Let $f(Z) = [f(z_1), ..., f(z_N)]^T$, $\circ$ denote the Hadamard product, $\mathbf{1}_{N \times N}$ a matrix of ones, and $I_N$ the identity matrix. Define $\mathcal{O} = \frac{1}{N-1}(\mathbf{1}_{NxN} - I_N) \circ (Z\mathcal{R}Z^T)$, and $\mathcal{D} = diag(\mathcal{O}\mathbf{1}_N)$,*

*to be a diagonal matrix. The CARMS estimator can equivalently be written in the following form:*

$$g_{CARMS}(\boldsymbol{Z}) = \frac{1}{N} f(\boldsymbol{Z})^T (\boldsymbol{\mathcal{D}} - \boldsymbol{\mathcal{O}}) \left(\boldsymbol{Z} - \boldsymbol{1}_N \sigma(\boldsymbol{\phi})^T\right). \tag{5}$$

*Furthermore, for independent samples, $\boldsymbol{Z}\boldsymbol{\mathcal{R}}\boldsymbol{Z}^T = \boldsymbol{1}_{N \times N}$ and LOORF has the form:*

$$g_{LOORF}(\boldsymbol{Z}) = \frac{1}{N} f(\boldsymbol{Z})^T \left(\boldsymbol{I}_{N \times N} - \frac{1}{N-1} \left(\boldsymbol{1}_{NxN} - \boldsymbol{I}_N\right)\right) \left(\boldsymbol{Z} - \boldsymbol{1}_N \sigma(\boldsymbol{\phi})^T\right) \tag{6}$$

### 3.1 Multivariate CARMS

In the univariate case, Monte Carlo estimators are not necessary, as the expectation has $C$ terms and can simply be analytically summed, but for many categorical variables, stochastic gradients are required. For $D$ dimensions, we can sample $\boldsymbol{Z}^d \sim \text{Cat}(\sigma(\boldsymbol{\phi}^d))$ independently and combine them into a $D \times N \times C$ tensor $\boldsymbol{Z}$, with the corresponding $D \times C \times C$ importance ratio tensor $\boldsymbol{\mathcal{R}}$. Below, we use superscripts and subscripts to index the first and second dimension of the tensors, with $\boldsymbol{Z}^{-d}$ denoting excluding the $d^{th}$ row of $\boldsymbol{Z}$. Focusing on the $d^{\text{th}}$ dimension, we have:

$$\nabla_{\boldsymbol{\phi}^d} \mathbb{E}\left[f(\boldsymbol{Z})\right] = \mathbb{E}_{\boldsymbol{Z}^{-d}} \left[\nabla_{\boldsymbol{\phi}^d} \mathbb{E}_{\boldsymbol{Z}^d} \left[f(\boldsymbol{Z}^{-d}, \boldsymbol{Z}^d)\right]\right]$$

$$= \mathbb{E}_{\boldsymbol{Z}^{-d}} \left[\mathbb{E}_{\boldsymbol{Z}^d} \left[\frac{1}{N(N-1)} \sum_{n \neq m} \frac{1}{2} \Big(f(\boldsymbol{Z}_n^{-d}, \boldsymbol{Z}_n^d) - f(\boldsymbol{Z}_m^{-d}, \boldsymbol{Z}_m^d)\Big)(\boldsymbol{Z}_n^d - \boldsymbol{Z}_m^d)\boldsymbol{Z}_n^{d^T}\boldsymbol{\mathcal{R}}^d \boldsymbol{Z}_m^d\right]\right]$$

$$= \mathbb{E}\left[\frac{1}{N(N-1)} \sum_{n \neq m} \frac{1}{2} \left(f(\boldsymbol{Z}_n) - f(\boldsymbol{Z}_m)\right) \left(\boldsymbol{Z}_n^d - \boldsymbol{Z}_m^d\right) \boldsymbol{Z}_n^{d^T}\boldsymbol{\mathcal{R}}^d \boldsymbol{Z}_m^d\right].$$

Just like the univariate case, we only need $N$ evaluations of $f$ regardless of the dimensionality $D$.

### 3.2 Antithetic categorical variables

To use CARMS in practice, we describe two ways to generate antithetic categorical variables in this section. Both methods are based on transformations of uniform random variables, which makes them amenable to antithetic copulas, such as the Gaussian or Dirichlet copula [Dimitriev and Zhou, 2021].

#### 3.2.1 Inverse CDF sampling

The inverse transform sampling is commonly used to transform uniform random variables, which are easy to generate, to some other distribution:

$$x \sim F_x(x) \iff u \sim \text{Unif}(0, 1), \ x = F_x^{-1}(u),$$

where $F_x(x)$ denotes the CDF of the desired distribution. For a categorical variable and some ordering of its probability vector $\boldsymbol{p} = (p_1, ..., p_C)$ the inverse CDF approach amounts to the following algorithm. Define the left and right boundaries:

$$\boldsymbol{l}_i = \sum_{k=1}^{i-1} p_k, \quad \boldsymbol{r}_i = \sum_{k=1}^{i} p_k = \boldsymbol{l}_i + \boldsymbol{p}_i. \tag{7}$$

Then setting $\boldsymbol{z} = \hat{\boldsymbol{j}}$ if $u \in [\boldsymbol{l}_j, \boldsymbol{r}_j]$, where $u \sim \text{Unif}(0, 1)$ produces a categorical variable because $P(\boldsymbol{z} = \hat{\boldsymbol{j}}) = P(u \in [\boldsymbol{l}_j, \boldsymbol{r}_j]) = \boldsymbol{r}_j - \boldsymbol{l}_j = p_j$. To obtain $N$ antithetic categorical variables, we sample $\boldsymbol{u} \sim \mathcal{C}_N$ from a copula and set $z_n$ in a vectorized manner. Importantly, this sampling approach allows us to easily analytically evaluate the bivariate PMF (proof in the Appendix):

$$P(\boldsymbol{z} = \hat{\boldsymbol{i}}, \boldsymbol{z}' = \hat{\boldsymbol{j}}) = \Phi_{\mathcal{C}}(\boldsymbol{r}_i, \boldsymbol{r}_j) - \Phi_{\mathcal{C}}(\boldsymbol{r}_i, \boldsymbol{l}_j) - \Phi_{\mathcal{C}}(\boldsymbol{l}_i, \boldsymbol{r}_j) + \Phi_{\mathcal{C}}(\boldsymbol{l}_i, \boldsymbol{l}_j), \tag{8}$$

where $\Phi_{\mathcal{C}}$ denotes the bivariate CDF of the copula. However, we are not guaranteed a non-zero probability for any $(i, j)$ pair within a particular ordering. For example, if the ordering is $\boldsymbol{p} = (0.1, 0.2, 0.7)$ and we use a bivariate copula $(u, 1 - u)$, then the probability of obtaining the pair (1,2) is zero:

$$P(\boldsymbol{z} = 1, \boldsymbol{z}' = 2) = P(u \in [0, 0.1], 1 - u \in [0.1, 0.3]) = P(u \in [0, 0.1], u \in [0.7, 0.9]) = 0.$$

---
**Algorithm 1** Antithetic inverse CDF categorical sampling
---
**Input:** Number of samples $N$, probabilities $\boldsymbol{p} = \sigma(\boldsymbol{\phi})$, copula $\mathcal{C}$.
Sample $\boldsymbol{u} = (u_1, ..., u_N) \sim \mathcal{C}_N$.
Shuffle $\boldsymbol{p}$ according to Eq. 9, and define the left and right boundaries $\boldsymbol{l}, \boldsymbol{r}$ according to Eq. 7.
Set $\forall n: \boldsymbol{z}_n = \hat{\boldsymbol{j}}$, where $j$ is such that $u_n \in [\boldsymbol{l}_j, \boldsymbol{r}_j]$.
For $i, j \in \{1, ..., C\}$: compute $\mathcal{R}_{ij} = \boldsymbol{p}_i \boldsymbol{p}_j / P(\boldsymbol{z}_n = \hat{\boldsymbol{i}}, \boldsymbol{z}_{n'} = \hat{\boldsymbol{j}})$ according to Eq. 10.
**return:** $(\boldsymbol{z}_1, ..., \boldsymbol{z}_N)$, $\mathcal{R}$.
---

---
**Algorithm 2** Antithetic Gumbel categorical sampling
---
**Input:** Number of samples $N$, probabilities $\boldsymbol{p} = \sigma(\boldsymbol{\phi})$, copula $\mathcal{C}$.
Sample $\boldsymbol{u}_c = (u_{c1}, ..., u_{cN}) \overset{iid}{\sim} \mathcal{C}_N$ for each category $c$.
Set $\forall c, n : g_{cn} = -\ln(-\ln(u_{cn}))$.
Convert to categorical $\forall n: \boldsymbol{z}_{nc} = 1$ if $c = \text{argmax}_d\, g_{dn}$ else 0.
Approximate the bivariate PMF $P(\boldsymbol{z} = \hat{\boldsymbol{i}}, \boldsymbol{z}' = \hat{\boldsymbol{j}})$ according to Eq 11.
Set $\forall i, j \in \{1, ..., C\}$: $\mathcal{R}_{ij} = \boldsymbol{p}_i \boldsymbol{p}_j / P(\boldsymbol{z} = \hat{\boldsymbol{i}}, \boldsymbol{z}' = \hat{\boldsymbol{j}})$.
**return:** $(\boldsymbol{z}_1, ..., \boldsymbol{z}_N)$, $\mathcal{R}$.
---

However, we are always guaranteed a non-zero probability for the pair (1, C) (proof for the Dirichlet copula in the appendix). Therefore, we can average over $C(C-1)/2$ orderings of $\boldsymbol{p}$, where the ordering $\sigma_{ij}$ rearranges $\boldsymbol{p}$ such that categories $i$ and $j$ are the first and last category of the vector, i.e. $\sigma_{ij}(i) = 1, \sigma_{ij}(j) = C$. Thus, any pair of categories $(i, j)$ has non-zero probability in at least one ordering, namely the ordering $\sigma_{ij}$. There are many sets of orderings that satisfy this, but one simple way to do this is to translate the original vector $\boldsymbol{p}$ $i$ elements to the right, then swap the last element with the $j^{\text{th}}$, or more precisely:

$$\sigma_{ij}(k) = \begin{cases} C, & k = j \\ j, & k = C \\ (k - i + 1) \bmod C, & \text{otherwise} \end{cases}. \tag{9}$$

The bivariate PMF given the set of orderings is:

$$P\left(\boldsymbol{z} = \hat{\boldsymbol{i}}, \boldsymbol{z}' = \hat{\boldsymbol{j}}\right) = \tfrac{2}{C(C-1)} \sum_{k<l} P\left(\boldsymbol{z} = \sigma_{kl}(\hat{\boldsymbol{i}}), \boldsymbol{z}' = \sigma_{kl}(\hat{\boldsymbol{j}}) \mid \sigma_{kl}(\boldsymbol{p})\right)$$

$$\approx \frac{1}{|O|} \sum_{o \in O} P\left(\boldsymbol{z} = \sigma_o(\hat{\boldsymbol{i}}), \boldsymbol{z}' = \sigma_o(\hat{\boldsymbol{j}}) \mid \sigma_o(\boldsymbol{p})\right). \tag{10}$$

If the number of categories $C$ is large, we can approximate the sum by sampling a random subset of orderings $O, |O| \ll C(C-1)/2$, as shown above. Note that we don't need to calculate Eq. 10 for all possible pairs of categories, just those that were in a sample of $N$. This results in $O(N^2|O|)$ total computation per gradient, and is easily parallelizable. We summarize the inverse CDF sampling method in Algorithm 1.

### 3.2.2 Gumbel max sampling

It is not strictly necessary to analytically calculate the bivariate PMF. If the variance is not too large, a Monte Carlo estimation suffices, and we can use the well known Gumbel max trick [Gumbel, 1954, Jang et al., 2017, Maddison et al., 2017]:

$$\boldsymbol{z} \sim \text{Cat}(\sigma(\boldsymbol{\theta})) \quad \Longleftrightarrow \quad g_c \overset{iid}{\sim} \text{Gumbel}(0, 1), \quad z_d = \begin{cases} 1, & d = \text{argmax}_c\, g_c + \theta_c \\ 0, & \text{otherwise} \end{cases}$$

where, as before, $\boldsymbol{z}$ denotes the one hot encoding of the categorical variable. This is also equivalent to the following exponential racing [Ross, 2014, Caron and Teh, 2012, Zhang and Zhou, 2018, Yin et al., 2019] sampling: set $\boldsymbol{z} = \hat{\boldsymbol{j}}$ iff $j = \text{argmin}_i \epsilon_i$, where $\epsilon_i \sim \text{Exp}(e^{\phi_i})$.

Since a Gumbel variable can be generated using a uniform: $g = -\ln(-\ln(u)), \ u \sim \text{Unif}(0, 1)$, we use a copula to generate $\boldsymbol{u}_c = (u_{c1}, ..., u_{cN})$. This produces a negative correlation between all pairs of samples for category $c$, and we do the same for all categories. If we let $\boldsymbol{Z} = [\boldsymbol{z}_1, .., \boldsymbol{z}_N]^T$ as

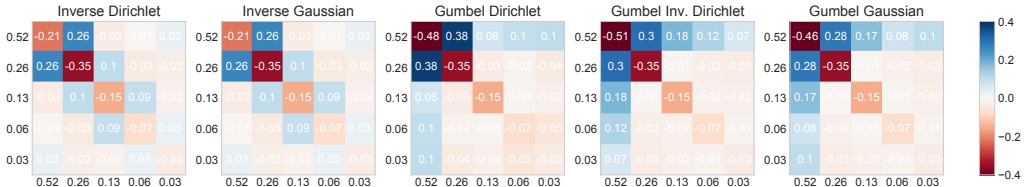

Figure 1: For an antithetic pair of one-hot encoded categorical variables $\boldsymbol{z}, \boldsymbol{z}'$ we show the correlation between each pair $\boldsymbol{z}_i$ and $\boldsymbol{z}'_j$, which are marginally Bernoulli variables with correlation $\rho_{ij} = \mathrm{corr}(\boldsymbol{z}_i, \boldsymbol{z}_j)$. The correlation matrices shown use different categorical sampling methods or different copulas, estimated using $N = 10^3$ Monte Carlo samples.

before, a simple empirical estimate of the bivariate PMF computed using all pairs and only matrix operations is:

$$P(\boldsymbol{z} = \hat{\boldsymbol{i}}, \boldsymbol{z}' = \hat{\boldsymbol{j}}) \approx \frac{1}{N(N-1)} \boldsymbol{Z}^T \left(\mathbf{1}_{N \mathrm{x} N} - I_N\right) \boldsymbol{Z}. \tag{11}$$

We summarize the antithetic Gumbel sampling method in Algorithm 2, and in Fig. 1 we show an example of a bivariate PMF produced by either the Gumbel or inverse CDF method using different copulas. The results are similar using either a Dirichlet or Gaussian copula, with larger differences between the categorical sampling method. For CARMS to be unbiased we need a good estimate of the ratio $p(z_i)p(z_j)/p(z_i, z_j)$, which in turn requires a good estimate of the bivariate probability mass function (PMF) in the denominator. If we draw $N$ samples to do this empirically, there is a small probability that some pair $(i, j)$ does not occur at all, so in experiments we clip the maximum value of the ratio to 10 to avoid infinities.

## 4 Experimental results

In this section, we first illustrate the variance reduction that CARMS offers on a toy example. We also optimize a categorical variational autoencoder (VAE) [Kingma and Welling, 2014, Rezende et al., 2014], and a stochastic network for structured output prediction, which are standard tasks [Jang et al., 2017] for categorical variables, done on three different benchmark datasets. Each experiment uses both antithetic categorical approaches for CARMS: inverse CDF sampling and the Gumbel max trick, denoted as CARMS-I and CARMS-G, respectively, and we use the Dirichlet copula for both. We compare our approach to three state-of-the-art unbiased estimators: LOORF/VarGrad [Kool et al., 2019a, Richter et al., 2020], the unordered set estimator (UNORD) [Kool et al., 2020], and ARSM [Yin et al., 2019]. The code for all experiments is freely available[2].

### 4.1 Toy example

We first showcase the variance reduction over other methods in a simple toy example, where we take the gradient with respect to the logits $\boldsymbol{\phi}$ of:

$$\mathcal{L}(\boldsymbol{\phi}) = \mathbb{E}[f(\boldsymbol{z}_1, \ldots, \boldsymbol{z}_D)], \quad \boldsymbol{z}_d \sim \mathrm{Cat}(\sigma(\boldsymbol{\phi_d})), \quad f(\boldsymbol{z}_1, \ldots, \boldsymbol{z}_D) = \sum_{d=1}^{D} \sum_{c=1}^{C} d \cdot c \cdot \boldsymbol{z}_{dc}.$$

For simplicity, let the number of categories, dimensions, and samples be $C = D = N = 3$. The probabilities are randomly sampled from a Dirichlet distribution: $\sigma(\boldsymbol{\phi}) \sim \mathrm{Dir}(\mathbf{1}_C \cdot \alpha)$, but are identical for all methods for a given $\alpha$. We vary the entropy of the probabilities from high ($\alpha = 1$) to low ($\alpha = 1000$). In a high entropy setting, there is little difference between the estimators, as the variance itself is very high, but differences emerge as we increase $\alpha$ and lower the entropy. The combined log variance of the gradient of each logit, for different methods and different $\alpha$, is shown in Fig 2.

---

[2]`https://github.com/alekdimi/carms`

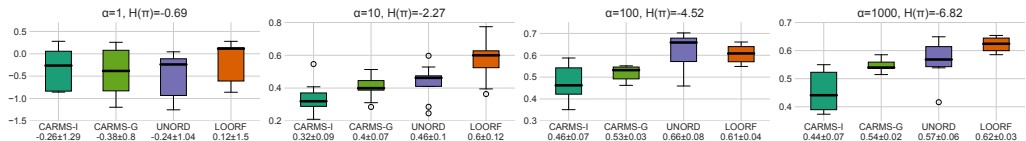

Figure 2: Log variance of the gradient of different estimators with respect to the logits on a toy problem. Columns correspond to different entropy levels of the logits.

## 4.2 Categorical variational autoencoder

For this task, we follow the experimental setting from ARMS [Dimitriev and Zhou, 2021], except we use categorical instead of binary latent variables, and maximize the ELBO:

$$\text{ELBO}(\boldsymbol{\phi}) = \mathbb{E}\Big[\ln\frac{p(\boldsymbol{x}|\boldsymbol{z})p(\boldsymbol{z})}{q_{\boldsymbol{\phi}}(\boldsymbol{z}|\boldsymbol{x})}\Big] \approx \sum_{n=1}^{N}\ln\frac{p(\boldsymbol{x}|\boldsymbol{z}_n)p(\boldsymbol{z}_n)}{q_{\boldsymbol{\phi}}(\boldsymbol{z}_n|\boldsymbol{x})}, \quad z_1,\dots,z_N \sim \text{Cat}_N(\sigma(\boldsymbol{\phi})),$$

where $\text{Cat}_N(\sigma(\boldsymbol{\phi}))$ denotes an $N$-variate categorical distribution with identical marginals. The number of categories is $C \in \{3, 5, 10\}$ with $D = \lfloor 200/C \rfloor$ latent variables, respectively, to make the total computational effort similar. In the binary case $C = 2$, CARMS reduces to ARMS, for which a thorough comparison has already been produced. The task is training a categorical VAE using either a linear or nonlinear encoder/decoder pair on three different datasets: Dynamic(ally binarized) MNIST [LeCun et al., 2010], Fashion MNIST [Xiao et al., 2017], and Omniglot [Lake et al., 2015]. All datasets are freely available under the MIT license, and do not contain any personally identifiable information or offensive content. For a fair comparison, all methods use the same learning rate, optimizer, model architecture, and number of samples. Since ARSM uses a variable number of function evaluations per step, we use one sample per step, for which ARSM uses around twice as many evaluations as the other methods. The results are combined from five independent runs for each experimental configuration.

The VAE consists of a stochastic layer with $\lfloor 200/C \rfloor$ units, each of which is a $C$-way categorical variable. For the nonlinear case, there are additionally two layers of 200 units with LeakyReLU [Maas et al., 2013] activations. The prior logits are optimized using SGD with a learning rate of $10^{-2}$, whereas the encoder and decoder are optimized using Adam [Kingma and Ba, 2015] with a learning rate of $10^{-4}$, following Yin et al. [2019]. The optimization is run for $10^6$ steps, with a batch size of 50, from which the global dataset mean is subtracted. The models are trained on an Intel Xeon Platinum 8280 2.7GHz CPU, and an individual run takes approximately 16 hours on one core of the machine, with a total carbon emissions estimated to be 28.19 kg of $CO_2$ [Lacoste et al., 2019]. An exception is UNORD for 10 samples, which was significantly more heavy in computation. Although the paper states that a step can be performed in $O(2^C)$, the provided code requires $O(C!)$ evaluations per step, forcing us to use $N = 8$ samples at most.

In Fig 3, we plot the training and test log likelihood using 100 samples, and gradient variance w.r.t the logits over time, for a nonlinear network on dynamic MNIST. Similar plots for other datasets and network types can be found in the Appendix. Also shown in Table 1 is the final training log likelihood using 100 samples for all three datasets, network types, categories, and gradient estimators. The corresponding table with the final test log likelihood can be found in the Appendix. In general, both versions of CARMS perform comparably, and result in slightly higher log likelihood than other methods. Because Gumbel CARMS uses an empirical estimate of the debiasing ratio, it has higher variance and slightly worse performance. When limiting the number of function evaluations, there is a gap between ARSM (which used on average $2C$ evaluations per step, out of a maximum of $C(C-1)/2$ per step) compared to the other methods, which used $C$ evaluations. This is possibly due to the continuous reparameterization that ARSM uses, which adds variance.

## 4.3 Structured prediction with stochastic categorical networks

We also compare all methods on the standard benchmark task of predicting the lower half of an image from the upper half, $i.e.$, the conditional distribution $p(\boldsymbol{x}_l \,|\, \boldsymbol{x}_u)$, where $\boldsymbol{x}_u$ and $\boldsymbol{x}_l$ denote the upper and lower half of an image, respectively. We use a stochastic categorical network to estimate this

Table 1: Final training 100 sample log likelihood of VAEs using different estimators, where the stochastic layer contains C=3, 5, or 10 categories, with $\lfloor 200/C \rfloor$ latent variables and $C$ samples per gradient step, respectively. Results are reported on three datasets: Dynamic MNIST, Fashion MNIST, and Omniglot over 5 runs, with the best performing methods in bold.

| | Categories | CARMS-I | CARMS-G | LOORF | UNORD | ARSM |
|---|---|---|---|---|---|---|
| **Dynamic MNIST** Linear | 3 | **-105.34 ± 0.25** | **-105.36 ± 0.24** | -105.64 ± 0.23 | **-105.23 ± 0.23** | -107.35 ± 0.56 |
| | 5 | **-103.53 ± 0.13** | **-103.35 ± 0.18** | -103.54 ± 0.15 | **-103.50 ± 0.13** | -106.13 ± 0.53 |
| | 10 | -103.22 ± 0.05 | **-103.12 ± 0.06** | -103.48 ± 0.06 | -103.56 ± 0.06 | -106.71 ± 0.58 |
| Nonlinr | 3 | **-94.85 ± 0.28** | **-94.60 ± 0.28** | -95.12 ± 0.21 | -95.21 ± 0.22 | -99.62 ± 0.50 |
| | 5 | -93.05 ± 0.14 | **-92.60 ± 0.12** | -92.91 ± 0.16 | -92.98 ± 0.12 | -98.89 ± 0.43 |
| | 10 | **-92.13 ± 0.05** | -92.42 ± 0.10 | -92.44 ± 0.04 | -93.05 ± 0.14 | -97.76 ± 0.41 |
| **Fashion MNIST** Linear | 3 | **-245.44 ± 0.22** | -245.69 ± 0.19 | -245.8 ± 0.19 | -245.90 ± 0.21 | -247.51 ± 0.45 |
| | 5 | -242.06 ± 0.13 | **-241.90 ± 0.10** | -242.17 ± 0.09 | -242.43 ± 0.12 | -244.63 ± 0.47 |
| | 10 | **-240.44 ± 0.03** | -240.52 ± 0.04 | -240.91 ± 0.04 | -241.08 ± 0.06 | -243.29 ± 0.36 |
| Nonlinr | 3 | **-233.13 ± 0.16** | **-233.20 ± 0.16** | -233.75 ± 0.10 | -233.34 ± 0.12 | -237.93 ± 0.20 |
| | 5 | **-231.72 ± 0.09** | **-231.67 ± 0.13** | -232.01 ± 0.05 | -232.19 ± 0.09 | -237.29 ± 0.34 |
| | 10 | **-230.77 ± 0.05** | -231.16 ± 0.05 | -231.35 ± 0.03 | -231.74 ± 0.02 | -235.88 ± 0.17 |
| **Omniglot** Linear | 3 | **-114.53 ± 0.12** | -114.73 ± 0.15 | -114.90 ± 0.13 | -114.77 ± 0.15 | -116.34 ± 0.42 |
| | 5 | **-114.01 ± 0.10** | **-113.93 ± 0.11** | **-114.01 ± 0.10** | **-114.00 ± 0.09** | -115.72 ± 0.35 |
| | 10 | **-114.97 ± 0.06** | **-114.99 ± 0.06** | -115.17 ± 0.05 | -115.55 ± 0.08 | -117.48 ± 0.35 |
| Nonlinr | 3 | **-110.19 ± 0.23** | **-110.16 ± 0.21** | **-110.33 ± 0.27** | **-110.31 ± 0.21** | -115.13 ± 0.51 |
| | 5 | **-109.14 ± 0.09** | -109.35 ± 0.13 | -109.39 ± 0.16 | -109.55 ± 0.14 | -115.07 ± 0.37 |
| | 10 | **-108.66 ± 0.08** | **-108.62 ± 0.07** | -108.98 ± 0.06 | -109.54 ± 0.15 | -115.11 ± 0.34 |

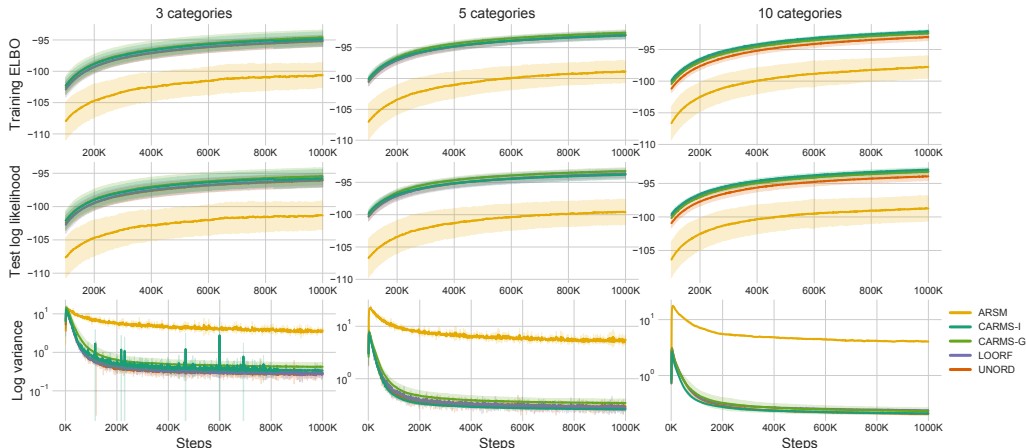

Figure 3: Training a nonlinear categorical VAE with different estimators on Dynamic MNIST using ELBO. Columns correspond to $C \in \{3, 5, 10\}$ categories with $C$ samples per gradient step, respectively. Rows correspond to the 100 sample training and test log likelihood, and the variance of the gradient with respect to the logits of the encoder network. Results for different datasets and other networks can be found in the Appendix.

distribution, with the objective: $\mathbb{E}_{\boldsymbol{z} \sim p(\boldsymbol{z}_m \mid \boldsymbol{x}_u)} \left[ \frac{1}{M} \sum_{m=1}^{M} \ln p(\boldsymbol{x}_l \mid \boldsymbol{z}_m) \right]$, where $\boldsymbol{z}$ denotes a stochastic categorical layer. The encoder/decoder pair each contains one hidden layer with $\lfloor 200/C \rfloor$ latent variables and a LeakyReLU activation, with the optimization performed for $C \in \{3, 5, 10\}$ categories, on all three datasets, with identical settings for each gradient estimator for a fair comparison. We use $M = 1$ for training, and $M = 1000$ for evaluation on both the train and test set. In Table 2, we show the final training set log likelihood, with the corresponding test log likelihood table in the appendix, though the results are qualitatively similar. The results are similar to the VAE experiment, with the inverse CDF CARMS having a slightly higher log likelihood. However, the differences between estimators are less pronounced, with the unordered set estimator is being mostly on par with CARMS, and ARSM only slightly trailing the other methods, with a much smaller gap.

Table 2: Final training log likelihood of a categorical network for conditional estimation using different gradient estimators, where the stochastic layer contains $C = 3$, 5, or 10 categories, with $\lfloor 200/C \rfloor$ latent variables and $C$ samples per gradient step, respectively. Results are reported on three datasets: Dynamic MNIST, Fashion MNIST, and Omniglot over 5 runs, with the best performing methods in bold.

| | Categories | CARMS-I | CARMS-G | LOORF | UNORD | ARSM |
|---|---|---|---|---|---|---|
| **Dynamic MNIST** | 3 | **57.98 ± 0.10** | 58.35 ± 0.14 | 58.19 ± 0.14 | **58.06 ± 0.04** | 60.22 ± 0.19 |
| | 5 | **57.57 ± 0.05** | 57.85 ± 0.12 | 57.78 ± 0.08 | **57.6 ± 0.05** | 59.38 ± 0.13 |
| | 10 | **58.17 ± 0.26** | **58.33 ± 0.13** | **58.20 ± 0.14** | **58.18 ± 0.17** | 59.32 ± 0.11 |
| **Fashion MNIST** | 3 | **132.83 ± 0.08** | **132.90 ± 0.08** | 133.10 ± 0.05 | 133.06 ± 0.06 | 134.56 ± 0.35 |
| | 5 | **132.68 ± 0.05** | 132.81 ± 0.12 | 132.91 ± 0.07 | 132.94 ± 0.14 | 134.09 ± 0.12 |
| | 10 | **133.32 ± 0.15** | **133.43 ± 0.23** | 133.54 ± 0.17 | **133.38 ± 0.10** | 134.02 ± 0.21 |
| **Omniglot** | 3 | **65.57 ± 0.10** | 66.05 ± 0.14 | 65.92 ± 0.26 | 65.81 ± 0.09 | 68.00 ± 0.09 |
| | 5 | **65.65 ± 0.18** | 66.16 ± 0.32 | 65.92 ± 0.23 | **65.78 ± 0.06** | 67.99 ± 0.32 |
| | 10 | **66.76 ± 0.31** | 66.94 ± 0.24 | **66.87 ± 0.07** | **66.66 ± 0.24** | 68.35 ± 0.16 |

## 5   Discussion

We have presented a novel approach for training categorical variables, which extends the ARMS estimator for the binary case. It goes beyond i.i.d. samples by using a copula to generate antithetic categorical samples, but preserves unbiasedness by including a multiplicative term. For i.i.d. samples, the form of the estimator reduces to LOORF. We showcase its usefulness on several datasets, a different number of categories and types of deep neural networks. In variational inference tasks, and conditional estimation, CARMS outperforms other state of the art estimators. The main limitation of this work is its specificity to categorical (including binary) variables. We hope to extend this, e.g. to Plackett-Luce models for top-k sampling [Kool et al., 2019b, Grover et al., 2019], which has important applications in ranking [Dadaneh et al., 2020]. There is also a general limitation that CARMS shares with other state-of-the-art estimators, which is higher complexity than LOORF, a very simple but strong baseline. Future work includes investigating theoretical properties and large scale applications, as well as possible general antithetic gradient estimators, and we plan to investigate adaptive correlations that take into account the properties of $f$ to further reduce the variance.

**Potential societal impact**   This work is focused on better optimization for categorical variables, which includes any network containing a stochastic categorical layer. In particular, generative models such as VAEs are widely used, and are sometimes trained on more human centered datasets. These models can sometimes be used to impersonate a person's face or voice. We only used non-human datasets such as MNIST, but with enough knowledge and using the available code, anyone, including bad actors, can train the same model on human content for malicious purposes. However, we strongly believe that wide knowledge dissemination and open source code is crucial for reproducibility in all of science.

## Acknowledgments

The authors acknowledge the support of NSF IIS-1812699, the APX 2019 project sponsored by the Office of the Vice President for Research at The University of Texas at Austin, the support of a gift fund from ByteDance Inc., and the Texas Advanced Computing Center (TACC) for providing HPC resources that have contributed to the research results reported within this paper.

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
