# Appendix for CARMS: Categorical-Antithetic-REINFORCE Multi-Sample Gradient Estimator

## A Background

### A.1 Copulas

Although any multivariate random variable with uniform marginal distributions is commonly referred to as a copula, strictly speaking, a copula is specifically its CDF. We are interested in copulas with strong negative dependence between any pair of its variables, which we refer to as antithetic in this work. They are also referred to as countermonotonic copulas when investigated in other work [Lee and Ahn, 2014, McNeil and Nešlehová, 2009]. However, only in the bivariate case do these copulas achieve the theoretical lower bound for negative dependence, the Fréchet–Hoeffding bound:

$$\mathcal{C}(u_1, ..., u_N) \geq \max\left\{1 - N + \sum_{i=1}^{N} u_i, 0\right\}.$$

Furthermore, it has been shown [McNeil and Nešlehová, 2009] that no copula can achieve this lower bound for more than two dimensions. However, the Dirichlet copula used in this work is in the family of Archimedean copulas, and Lee and Ahn [2014] show that within this family, the Dirichlet copula has the strongest countermonotonicity. Its CDF is defined as:

$$\mathcal{C}(u_1, ..., u_N) = \left(\max\left\{0, u_1^{\frac{1}{N-1}} + ... + u_N^{\frac{1}{N-1}} - (N-1)\right\}\right)^{N-1}.$$

### A.2 Copula sampling

We use the Dirichlet copula described in ARMS [Dimitriev and Zhou, 2021], which transforms a Dirichlet vector with concentration $\boldsymbol{\alpha} = \mathbf{1}$ to a copula sample. Besides its countermonotonic properties, we choose this copula because both its univariate and bivariate marginal CDFs are analytically tractable. The former is required for transforming the Dirichlet vector $\boldsymbol{d} \sim \text{Dir}(\boldsymbol{\alpha})$ into uniform variables: $\boldsymbol{u} = 1 - (1 - \boldsymbol{d})^{N-1}$. For Eq. 8, we want to analytically calculate the bivariate joint for categorical sampling. This requires the bivariate CDF, which has the form:

$$P(\boldsymbol{u}_i < p, \boldsymbol{u}_j < q) = P(\boldsymbol{d}_i < 1 - (1-p)^{1/(N-1)}, \boldsymbol{d}_j < 1 - (1-q)^{1/(N-1)})$$
$$= p + q - 1 + \max(0, (1-p)^{1/(N-1)} + (1-q)^{1/(N-1)}$$

### A.3 Pair equivalent definition of LOORF

Because the LOORF for $N$ samples for any distribution can be decomposed into all pairs, we can reuse the theorem, and we reproduce the short algebra needed to show it below:

$$\begin{aligned}
g_{\text{loorf}}(\boldsymbol{b}) &= \frac{1}{n} \sum_{i=1}^{n} \left(f(\boldsymbol{b}_i) - \frac{1}{n-1} \sum_{j \neq i} f(\boldsymbol{b}_j)\right) \nabla_\phi \ln p(\boldsymbol{b}_i) \\
&= \frac{1}{n} \sum_{i=1}^{n} f(\boldsymbol{b}_i) \nabla_\phi \ln p(\boldsymbol{b}_i) - \frac{1}{n(n-1)} \sum_{i=1}^{n} \nabla_\phi \ln p(\boldsymbol{b}_i) \sum_{j \neq i} f(\boldsymbol{b}_j) \\
&= \frac{1}{n(n-1)} \sum_{i \neq j} \left(f(\boldsymbol{b}_i) \nabla_\phi \ln p(\boldsymbol{b}_i) - f(\boldsymbol{b}_i) \nabla_\phi \ln p(\boldsymbol{b}_j)\right) \\
&= \frac{1}{n(n-1)} \sum_{i \neq j} \frac{1}{2} (f(\boldsymbol{b}_i) - f(\boldsymbol{b}_j))(\nabla_\phi \ln p(\boldsymbol{b}_i) - \nabla_\phi \ln p(\boldsymbol{b}_j)) \\
&= \frac{1}{n(n-1)} \sum_{i \neq j} g_{\text{pod}}(b_i, b_j).
\end{aligned}$$

# B Proofs

## B.1 Proof of Theorem 2

Define $\Delta f_{ij} = f(\boldsymbol{z}_i) - f(\boldsymbol{z}_j)$. Using the assumption that $P(\boldsymbol{z} = \hat{\boldsymbol{\imath}}, \boldsymbol{z}' = \hat{\boldsymbol{\jmath}}) \geq P(\boldsymbol{z} = \hat{\boldsymbol{\imath}})P(\boldsymbol{z}' = \hat{\boldsymbol{\jmath}})$, and starting with CARTS we have:

$$
\begin{aligned}
\mathrm{Var}[g_{\mathrm{CARTS}}] &= \sum_{i \neq j} P(\boldsymbol{z} = \hat{\boldsymbol{\imath}}, \boldsymbol{z}' = \hat{\boldsymbol{\jmath}}) \left( \frac{1}{2} \Delta f_{ij}(\hat{\boldsymbol{\imath}} - \hat{\boldsymbol{\jmath}}) \frac{P(\boldsymbol{z} = \hat{\boldsymbol{\imath}})P(\boldsymbol{z}' = \hat{\boldsymbol{\jmath}})}{P(\boldsymbol{z} = \hat{\boldsymbol{\imath}}, \boldsymbol{z}' = \hat{\boldsymbol{\jmath}})} \right)^2 \\
&= \sum_{i \neq j} P(\boldsymbol{z} = \hat{\boldsymbol{\imath}})P(\boldsymbol{z}' = \hat{\boldsymbol{\jmath}}) \left( \frac{1}{2} \Delta f_{ij}(\hat{\boldsymbol{\imath}} - \hat{\boldsymbol{\jmath}}) \right)^2 \frac{P(\boldsymbol{z} = \hat{\boldsymbol{\imath}})P(\boldsymbol{z}' = \hat{\boldsymbol{\jmath}})}{P(\boldsymbol{z} = \hat{\boldsymbol{\imath}}, \boldsymbol{z}' = \hat{\boldsymbol{\jmath}})} \\
&< \sum_{i \neq j} P(\boldsymbol{z} = \hat{\boldsymbol{\imath}})P(\boldsymbol{z}' = \hat{\boldsymbol{\jmath}}) \left( \frac{1}{2} \Delta f_{ij}(\hat{\boldsymbol{\imath}} - \hat{\boldsymbol{\jmath}}) \right)^2 = \mathrm{Var}[g_{\mathrm{LOORF}}]
\end{aligned}
$$

Below is an example that satisfies the conditions. Let $\boldsymbol{p} = (0.6, 0.3, 0.1)$, for which we show the independent vs one possible antithetic pmf:

$$
\mathrm{pmf}_{\mathrm{indep}} = \boldsymbol{p}\boldsymbol{p}^T = \begin{bmatrix} 0.36 & 0.18 & 0.06 \\ 0.18 & 0.09 & 0.03 \\ 0.06 & 0.03 & 0.01 \end{bmatrix} \qquad \mathrm{pmf}_{\mathrm{anti}} = \begin{bmatrix} 0.30 & 0.24 & 0.06 \\ 0.24 & 0.02 & 0.04 \\ 0.06 & 0.04 & 0.01 \end{bmatrix}.
$$

Because every off-diagonal element of $\mathrm{pmf}_{\mathrm{anti}}$ is larger or equal to the corresponding independent pmf value, the variance is guaranteed to be no larger:

$$
\begin{aligned}
\mathrm{Var}(g_{\mathrm{CARTS}}) &= 2 \left( 0.24 \Delta f_{12}[1, 1, 0]^T \left( \frac{0.18}{0.24} \right)^2 + 0.06 \Delta f_{13}[1, 0, 1]^T + 0.04 \Delta f_{23}[0, 1, 1]^T \left( \frac{0.03}{0.04} \right)^2 \right) \\
&= 2 \left( 0.18 \Delta f_{12}[1, 1, 0]^T \frac{0.18}{0.24} + 0.06 \Delta f_{13}[1, 0, 1]^T + 0.03 \Delta f_{23}[0, 1, 1]^T \frac{0.03}{0.04} \right) \\
&\leq 2 \left( 0.18 \Delta f_{12}[1, 1, 0]^T + 0.06 \Delta f_{13}[1, 0, 1]^T + 0.03 \Delta f_{23}[0, 1, 1]^T \right) = \mathrm{Var}(g_{\mathrm{2LOORF}}).
\end{aligned}
$$

## B.2 Proof of Lemma 4

First, note that the $(ij)^{\mathrm{th}}$ element of $\boldsymbol{\mathcal{D}} - \boldsymbol{\mathcal{O}}$ is $\left( \sum_{j \neq i} \boldsymbol{\mathcal{R}}_{ij} \right) - \boldsymbol{\mathcal{R}}_{ij}$. Then we can explicitly write out the summation, and distribute it into all pairs, to arrive at the form of Theorem 3:

$$
\begin{aligned}
g_{\mathrm{CARMS}} &= \frac{1}{N} f(\boldsymbol{Z})^T (\boldsymbol{\mathcal{D}} - \boldsymbol{\mathcal{O}}) \left( \boldsymbol{Z} - \boldsymbol{1}_N \sigma(\boldsymbol{\phi})^T \right) \\
&= \frac{1}{N} \sum_{i=1}^{N} f(\boldsymbol{Z})_i^T \left( \sum_{j \neq i} \boldsymbol{\mathcal{R}}_{ij} \right) - \left( \sum_{j \neq i} f(\boldsymbol{Z})_j^T \boldsymbol{\mathcal{R}}_{ij} \right) (\boldsymbol{Z}_i - \sigma(\boldsymbol{\phi})) \\
&= \frac{1}{N(N-1)} \sum_{i \neq j} \left( f(\boldsymbol{Z})_i - f(\boldsymbol{Z})_j \right) (\boldsymbol{z}_i - \boldsymbol{z}_j) \boldsymbol{z}_i^T \boldsymbol{\mathcal{R}}_{ij} \boldsymbol{z}_j
\end{aligned}
$$

## B.3 Proof of Equation 8

$$
\begin{aligned}
P(\boldsymbol{z} = \hat{\boldsymbol{\imath}}, \boldsymbol{z}' = \hat{\boldsymbol{\jmath}}) &= P\left( u \in [\boldsymbol{l}_i, \boldsymbol{r}_i], u' \in [\boldsymbol{l}_j, \boldsymbol{r}_j] \right) \\
&= P\left( u \leq \boldsymbol{r}_i, u' \in [\boldsymbol{l}_j, \boldsymbol{r}_j] \right) - P\left( u \leq \boldsymbol{l}_i, u' \in [\boldsymbol{l}_j, \boldsymbol{r}_j] \right) \\
&= P\left( u \leq \boldsymbol{r}_i, u' \leq \boldsymbol{r}_j \right) - P\left( u \leq \boldsymbol{r}_i, u' \leq \boldsymbol{l}_j \right) \\
&\qquad - P\left( u \leq \boldsymbol{l}_i, u' \leq \boldsymbol{r}_j \right) + P\left( u \leq \boldsymbol{l}_i, u' \leq \boldsymbol{l}_j \right) \\
&= \Phi_{\mathcal{C}}(\boldsymbol{r}_i, \boldsymbol{r}_j) - \Phi_{\mathcal{C}}(\boldsymbol{r}_i, \boldsymbol{l}_j) - \Phi_{\mathcal{C}}(\boldsymbol{l}_i, \boldsymbol{r}_j) + \Phi_{\mathcal{C}}(\boldsymbol{l}_i, \boldsymbol{l}_j).
\end{aligned}
$$

## B.4 Proof that the bivariate PMF for the pair (1, C) is non-zero

We show that joint probability of the first and last category $P(z = \hat{\mathbf{1}}, z' = \hat{C})$ is always positive, because the Dirichlet copula has an area of non-zero density in a subset of this region. If we take $\epsilon = \min(p_1, p_C) > 0$, then:

$$P(z = \hat{\mathbf{1}}, z' = \hat{C}) = P(u < p_1, u' > 1 - p_C) \geq P(u < \epsilon, u' > 1 - \epsilon)$$

$$= P(u < \epsilon) - P(u < \epsilon, u' < 1 - \epsilon) = \epsilon - \max\left\{0, \epsilon^{\frac{1}{N-1}} + (1 - \epsilon)^{\frac{1}{N-1}} - 1\right\}^{N-1}.$$

If the second term is zero, it trivially holds that the probability is non-zero since $\epsilon > 0$. Otherwise, note that for any integer $N > 1$:

$$\epsilon > 0 \iff 1 - \epsilon < 1 \iff (1 - \epsilon)^{\frac{1}{N-1}} < 1 \iff (1 - \epsilon)^{\frac{1}{N-1}} - 1 < 0$$

$$\iff \epsilon^{\frac{1}{N-1}} + (1 - \epsilon)^{\frac{1}{N-1}} - 1 < \epsilon^{\frac{1}{N-1}} \iff \left(\epsilon^{\frac{1}{N-1}} + (1 - \epsilon)^{\frac{1}{N-1}} - 1\right)^{N-1} < \epsilon$$

$$\iff \epsilon - \left(\epsilon^{\frac{1}{N-1}} + (1 - \epsilon)^{\frac{1}{N-1}} - 1\right)^{N-1} > 0,$$

which concludes the proof.

## C  Additional results