# OpenReview forum: "CARMS: Categorical-Antithetic-REINFORCE Multi-Sample Gradient Estimator"
_NeurIPS.cc/2021/Conference — NeurIPS 2021 Poster_

### Official Review · Reviewer_1mCY · 2021-07-16

**Rating:** 6
**Confidence:** 4

**Summary:**

This paper tackles the problem of differentiating expectations of discrete random variables. Specifically, this paper proposes a generalization of the recent ARMS approach to more than two categories. The CARMS gradient estimator uses multiple samples from a categorical distribution. Unlike e.g. LOORF, CARMS uses samples that are dependent (typically negatively correlated), hoping to further reduce estimator variance. The authors prove how to construct an unbiased gradient from these dependent samples using importance sampling. The authors propose several joint distributions (copulas) to actually generate such dependent samples. Experimental results are provided on several problems such as a categorical latent VAE.

**Limitations And Societal Impact:**

The authors have adequately addressed limitations and societal impact. For more details, see the main review.

**Main Review:**

Novelty: This work constitutes a natural extension of ARMS. To the best of my knowledge, this exact approach has not appeared before in the literature. This paper does a very good job of summarizing recent connected work in the field (Sections 1 and 2).

Quality: The core concept (Theorem 1) is well built up to and well explained. It’s clear where the estimator comes from and why it’s unbiased (from the preceding few lines). The approach and maths follows naturally from ARMS. Theorem 2 adds justification for the CARMS approach, the variance can be provably lower.

One thing that could make the paper better is a more thorough discussion of how to generate antithetic categorical variables (Sec 3.2), because this is more involved than in the 2-category case.
- I would like more details on the random reordering and the statement “Unfortunately, keeping to one specific ordering does not allow for all possible pairs to have non-zero probabilities”. It’s not clear to me why we cannot have distributions on [0,1]^2 which have positive density across the entire square, have uniform marginals and are not independent. Such a pair of variables, when discretized as per (8) would surely assign positive probability to every pair. Going further, can’t we transform the u, u’ distribution under a reordering by suitably shuffling the joint density between the different sub-rectangles of [0,1]^2? (Maybe this is something obvious that I missed, but it would be nice to have an explanation).
- How restrictive / how general is the inverse CDF method with uniforms? Is it possible that there exist more “natural” joint distributions for N dependent samples of a categorical?
- Is it clear why the Gumbel max approach seems to give more strongly negatively correlated samples than the Inverse CDF method? Or is this an artifact of the estimation of the joint distribution of the Gumbel method?

For both ARMS and CARMS, the benefit of antithetic sampling decreases as we add more samples. Does this decay happen more slowly for CARMS? For example, with a uniform categorical on C items we can make a coupling with N=C samples (e.g. by cycling one sample) that guarantees z_i != z_j. Discussion on this point might further add to the paper.

For the experiments, I think it’s good to consider a fair range of baselines, which the authors appear to have done. The results are not strong improvements over existing methods, but they do offer marginal improvements in most cases. The presentation of Fig 3 could be improved because it is impossible to visually disentangle most of the methods.

The computational complexity is mentioned in the discussion, but it could be worth explicitly quantifying it, e.g. in Sec 3.1 to mathematically explain the benefit of only using N evaluations of f we could write out the total cost of a gradient step of CARMS and compare to LOORF. This could also aid a theoretical comparison with UNORD and ARSM, as the intro suggests both are costlier than CARMS.

Clarity: the paper is mostly clear. Particularly earlier sections. I have made a few suggestions / queries about Sections 3.1 and 3.2 which would also benefit clarity. Overall, I find the exposition to be clear and helpful.

Significance: this paper touches on an interesting and relevant area of ML research: differentiating through categorical random variables. It forms a natural step in the logical progression of previous ideas (ARMS) and would therefore be a welcome and valuable component of the literature in this area. The fact that CARMS gets (slightly) better empirical results that previous methods also indicates that it can transfer to the practical setting well. On the other hand, the paper is natural next step with modest empirical improvements, rather than a seminal work.

Explanation of score
----------------------
I was torn between Accept and Weak Accept. Whilst I enjoyed reading the paper, the paper makes a modest step from ARMS, and has modest empirical results. It is great to have a more technically focused paper, but with that in mind, I think a bit more detail in Secs 3.1 and 3.2 would step things up to Accept for me.

Typos
-------
135: errant full stop

(7) : l_i^{(o)} should sum to i-1 not j-1


**Time Spent Reviewing:**

6

---

> ### Author Response · Authors · 2021-08-10
> **Response to reviewer 1mCY**
>
> We thank the reviewer for the thoughtful comments. First, we offer some intuition below on why is it not straightforward to sample antithetic categorical variables:
>
> - It is possible to have positive density across all of $[0, 1]^2$, but the goal is for this copula's marginals to be negatively correlated so that discretizing its marginals into categorical variables results in a negative correlation of each category. For example, let the pair be $(u, 1 - u)$, the best achievable negative correlation of the two uniform marginals. Let the 3 categories have probability $1/3$, so we discretize the lines into $[0, 1/3)$ for category $1$, $[1/3, 2/3)$ for category $2$ etc. Notice that when sampling $(u, 1 - u)$ we will either obtain the pair $(z=2, z'=2)$ if $u$ is between $1/3$ and $2/3$, or the pair $(z=1, z'=3)$ (or vice versa) otherwise. This means we have achieved excellent "antitheticness" for categories $1$ and $3$, as we will never obtain the duplicates $(1, 1)$ or $(3, 3)$. However, we also provide no information for category $2$, because there is zero probability of obtaining the pairs $(1, 2)$ and $(2, 3)$, so we cannot form an unbiased estimate of the expectation. This is simply due to the ratio $p(z=1)p(z'=2)/p(z=1, z'=2)$ being undefined, a basic requirement of importance sampling (non-zero density of the approximating distribution $q(x)$ wherever the original distribution $p(x) > 0$). We will expand on the complexity of our approach in the methods section.
>
> Regarding the different ways of sampling antithetic categorical variables:
>
> - Our experiments also included Gumbel sampling instead of the inverse CDF because we wanted to know whether performance is highly dependent on the sampling method, which does not seem to be the case. We do not think there is a more or less natural way to sample, but other categorical sampling approaches exist, such as binary decision trees or stickbreaking. We have further confirmed that the exact sampling method is not crucial in current separate work using a stickbreaking approach. We observed that which copula or sampling method gives better negative correlations seems to depend on the specific vector of probabilities themselves, with no discernible pattern.
>
> Regarding the diminishing benefits as we add more samples and complexity of the method:
>
> - In the ARMS toy experiment (Dimitriev and Zhou, 2021), it is shown that even for $N=100$ samples, there is markedly lower variance across any probability $p$ using antithetic sampling rather than independent. However, the reviewer raises a good point that in the categorical case here we have an issue of complexity. Luckily, the computation of CARMS scales not with the number of categories $C$, but with the number of unique pairs of samples, which can be up to $N(N-1)/2$ in the worst case. In practice, however, even when $C$ is large, $N \ll C$.
>
> - The cycling example with the coupling $N=C$ would actually not work even for the uniform probability example. Although the marginals would be correct, it does not produce non-zero probability for all pairs of different $z_i \neq z_j$, which is a required condition as mentioned above.
>
> - ARSM has a variable number of evaluations per step, up to $O(C^2)$, which can be huge, and requires all dimensions to have the same number of categories for the swaps. However we can directly control the number of evaluations and do not have this requirement. Furthermore, as mentioned above, CARMS is $O(N^2)$, whereas UNORD has complexity $O(2^N)$ in theory (Kool et al., 2020), and we find that the available implementation for UNORD is $O(N!)$, which resulted in difficulties when running $10$ category UNORD experiments. For $C=10$ we were forced to upper bound the number of samples to $N=8$, the highest number presented in Kool et al. (2020)'s experiments, which will be stated clearly.
>
> We will also fix any typos, and expand on Section 3.2 to make all of the above clearer.

---

> > ### Comment · Reviewer_1mCY · 2021-08-25
> > **Thank you for your reply**
> >
> > Thanks for a really detailed response to my review. I particularly appreciate the time taken to help me understand the requirements of the coupling. I think adding these kinds of details to the paper would strengthen the clarity and help the reader. I also appreciate the added discussion about complexity.
> >
> > I have decided to maintain my score, but I will be supporting this paper in the reviewer discussion.

---

> > > ### Author Response · Authors · 2021-08-30
> > > **Thank you for your support!**
> > >
> > > We sincerely appreciate your taking the time to read our response and providing additional feedback! We will add these details to the paper to strengthen the clarity.

---

### Official Review · Reviewer_eSUj · 2021-07-16

**Rating:** 5
**Confidence:** 4

**Summary:**

The paper proposes a categorical version of ARMS, namely CARMS.
By utilizing REINFORCE together with copula sampling, CARMS remains an unbiased estimator and non-duplicated samples reduce the variance of gradients.
The authors verify the proposed model through toy example, VAE, and the structured output prediction task.


**Limitations And Societal Impact:**

The authors pointed out the limitation, the complexity, of their work in one line in the conclusion.
However, my thought is that the complexity is somewhat crucial to the proposed method as the number of categories grows so that it should be dealt with in a separate subsection.
Meanwhile, the work seems that it does not have any potential negative societal impact.

**Main Review:**

Originality:
I do not think the natural extension from the previous idea harms the originality or the novelty of the paper "in general".
However, in this case, since CARMS is an extension of ARMS over the number of categories, I think the paper also needs to focus on the obstacle which follows from the natural extension which is discussed in the significance part.
Therefore, I will suspend the judgment on the originality until the authors' response.


Quality and Clarity:
The paper is grounded by theorems and the proofs are provided in the appendix.
It will be helpful for readers if the authors add a minimum introduction on the copula in the background section.
Also, the paper is written in a clear manner.


Significance:
- While the binary version (ARMS) may not suffer from the complexity, it can be a huge drawback for the proposed CARMS when the number of categories grows. However, there is no specific discussion about the complexity, nor the report on the running time in the experimental section. ARMS is a good estimator for the binary variable, however, CARMS, which is the natural extension of ARMS, seems that it needs to break through the complexity issue. Also, the performance gap is too marginal even though the proposed theorems are grounded. If the complexity really matters, why not use the comparable baselines? (which are simple and powerful) Or, CARMS needs to find a suitable application that CARMS outperforms other baselines with meaningful performance gaps.
- There are some missing baselines: RELAX, REBAR, etc.
- As a minor comment, the code is not available for reproducibility.


Questions:
- What does ECRAMS stand for?
- How does the experimental result change for the toy example if we sample the probabilities from Dir(\alpha) \alpha << 1? I'm wondering about the behavior of the proposed and baseline methods when the probability is extreme.
- Similarly, how does the performance change for the non-trivial or extreme prior in the VAE case?
- The complexity seems problematic. What is the exact computational complexity, and how does the complexity changes from ARMS to CARMS? (if there are the same number of latent dimensions) Also, could you compare the running times of the experiments? How does the running time change as the number of categories grows, for examples, C = 100, 200?


**Time Spent Reviewing:**

8

---

> ### Author Response · Authors · 2021-08-10
> **Response to reviewer eSUj**
>
> We thank the reviewer for the thoughtful comments and especially honing in on the one difficulty compared to the binary case. First, on the minor issues:
>
> - If space permits during the revision, we will add a short description of copulas in the main paper, otherwise will add it to the appendix.
>
> - We apologize for the lack of code in the supplementary material, it was a last minute oversight when uploading the supplement to OpenReview. The code will be made publicly available, with all the experiments readily reproducible.
>
> - Similarly, in a few places, CARMS-I and CARMS-G are erroneously labeled as CARMS and ECARMS (standing for empirical CARMS), which will also be fixed.
>
> - We ran experiments using RELAX (which generalizes REBAR), but they did not result in comparably good performance as reported in ARMS or DisARM. In addition, RELAX is not as fully general as the other methods in the experimental section. Thus, we decided to omit it and focus on other REINFORCE based estimators.
>
> - For $\alpha \ll 1$, the Dirichlet distribution produces sparse categorical probability vectors. We are not interested this case, since one category is overwhelmingly likely to be sampled, and the categorical variable is no longer random.
>
> - About non-uniform VAE priors: although the prior logits are initialized to $1/C$, they are learned throughout the VAE training, same as in the DisARM and ARM experiments.
>
> Regarding the main issue, which is the added complexity of moving to the categorical case:
>
> - The number of categories is less of a problem than at first glance, as crucially we are bounded by the number of samples N in each iteration, not the number of categories C. When C is large, we have $N << C$. In this case, there can only be up to N(N - 1)/2 different pairs of probabilities that need to be estimated. For example, if we draw categories 23, 45, 67, we only need to estimate the $3*(3-1)/2 = 3$ bivariate PMF values of the form $p(z = 23, z' = 45)$. Moreover, this is a worse case upper bound, as there will still be some amount of duplicate samples. In practice, $N$ is usually a small number, and we only used $N=C$ to reduce any researcher degrees of freedom. Thus, the complexity is proportional to the number of unique pairs $O(N^2)$ and not to the number of categories. Lastly, this calculation is easily vectorized.
>
> - We are considering adding an experiment to showcase this for a large number of categories, similarly to the toy example that shows the low entropy advantage.  We are also thoughtfully considering what exact experiment to add where reducing duplicate samples improves the performance by a larger margin. We agree that more time should be spent elaborating on it and will extend its discussion in the methods section.

---

> ### Author Response · Authors · 2021-08-24
> **Link to code and additional clarification on scalability**
>
> We have anonymized and uploaded the [code](https://mega.nz/file/iXognZzI#LnDdL8btaoX0GVtglIUdAhyFtW0ETWEE1h5RPiGtyQo) for reproducibility.
> (note according to NeurIPS 2021 Paper FAQ: “External links are discouraged, but you may include a link if it is required to respond to a question from a reviewer.”)
>
> Regarding the complexity issue, we would like to further clarify exactly how the inverse CDF approach is scalable and implemented in practice. Only step 3 needs to be adjusted for a large number of categories $C$.
>
> 1. As mentioned above, for $N$ samples, there are at most $N(N-1)/2$ pairs for which we need to calculate $p(z_i, z'_j)$, and $N$ is fully within our control during optimization.
>
> 2. Given an ordering of the logits, the above bivariate pmf requires four simple evaluations of the bivariate copula pdf. For the Dirichlet copula this is analytical and easily calculated as shown in Dimitriev and Zhou (2021).
>
> 3. The sum over all orderings can easily be approximated by a Monte Carlo approach, by setting a number $O \ll C^2$ and simply taking a small number of orderings $O$. The variance can be reduced by taking a non-uniform distribution over the orderings, and is quite flexible. For example, we can even use a Plackett-Luce distribution over all $C!$ orderings, where at each gradient step we take $O$ samples over which we average (Eq. 10).

---

### Official Review · Reviewer_qwuP · 2021-07-19

**Rating:** 6
**Confidence:** 3

**Summary:**

**Summary**
- The paper introduces a multi-sample gradient estimator for categorical variables.
- The estimator is based on the score-function and uses correlated samples to achieve a variance reduction.
- The estimator extends several recently proposed estimators naturally.
- The estimator yields marginal improvements over baselines in the experimental evaluations.

**Edit after Author Response**
- Thank you for replying to the points I raised. I think the changes you do improve the draft. However, because I find the experimental improvements fairly marginal, I have kept my score unchanged. I believe the paper is above the acceptance threshold.

**Ethical Concerns:**

N/A.

**Limitations And Societal Impact:**

N/A.

**Main Review:**

**Strengths**
- Originality. The contribution is a natural extension of existing work.
- Quality. To me, the contribution appears technically sound. The derivations and arguments are clear.
- Clarity. The paper is well-written. I enjoyed reading it and was able to follow the line of argument and the derivations.
- Significance. Gradient estimator for categorical distributions has wide-spread applications and draws interest from the community.

**Weaknesses**
- Quality/ Significance The improvements demonstrated in the experimental evaluation are marginal.

**Decision**
- Weak Accept. I think the paper is above the acceptance threshold, but I wished the experimental results were more convincing.

**Suggestions**

*Are duplicate samples really a significant issue?*
- Based on the paper, I suppose that the benefit of using this estimator instead of LOORF are most significant, when the duplicate sampling issue is acute. Based on the experiments presented, I am not convinced that duplicate samples are a significant issue in most practical applications.
- I would recommend to the authors to seek applications where the duplicate sampling issue is more acute to clearly demonstrate the utility of their estimator. Speculative: Maybe, as models converge, the entropy of the latent distributions shrinks. If this is the case, one could consider an experiment, where the estimators are used for fine-tuning a model which has been trained for significantly more iterations than the models presented in 4.2 (10^6 iterations).

*I am concerned about the statistical significance of the improvements.*
- The improvements appear very marginal, and I am not sure what the bolding scheme is indicating. Is this based on the standard errors, shout it actually be based on confidence intervals? I am unsure.
- Rather than bolding, would it be possible to carry out a hypothesis test (t-test) to test whether the means of the proposed method are significantly different from the baselines?

*Isn't it better to tune hyper-parameters exhaustively for each method separately?*
- I believe it is better to tune hyper-parameters individually for each method rather than choosing a fixed set of hyper-parameters on which all methods are compared against (which is what I understand, the paper did).
-It may be acceptable in this case though, because all methods are fairly similar anyway, and the optimal hyper-parameters might be similar/ the same as a result.

*What is ecarms in the plots?*
- The experimental set-up promised Carms-I and Carms-G. Maybe, I missed it, though.

*There is more related work on Rao-Blackwellization for gradient estimation for discrete distributions.*
- The authors survey related work on Rao-Blackwellization to improve gradient estimation, e.g. Titsias and Lazaro-Gredilla (2015) and Dong et al. (2021).  Rao-Blackwellization is increasngly being used for improving gradient estimators for discrete distributions and I think it's worth highlighting also other work in this area, e.g. Liu et al. (2018), Kool et al. (2020) or Paulus et al. (2021).

**References**
- Liu et al. (2018). Rao-Blackwellized stochastic gradients for disccrete distributions.
- Kool et al. (2020) Estimating gradients for discrete random variables by sampling without replacement.
- Paulus et al. (2021). Rao-Blackwellizng the Straight-Through Gumbel-Softmax estimator.

**Time Spent Reviewing:**

3

---

> ### Author Response · Authors · 2021-08-10
> **Response to reviewer qwuP**
>
> We thank the reviewer for these thoughtful comments, and respond to each point below.
>
> - We agree that the method has a meaningful advantage when duplicate samples arise. However, our approach is never worse than the independent case, as reducing duplicate samples can only serve to lower the variance and never increase it. The magnitude of the improvement will depend, as pointed out, by the entropy of the categorical distribution. In low entropy, when one or a few categories has a large probability, there is a large chance of duplicated samples. We will thoughtfully consider adding an experiment where we can showcase this case, such as late stage training or VAEs, and we thank the reviewer for their suggestion.
>
> - In the tables, for a specific row, we bolded any method whose ELBO (or log likelihood) was within one standard deviation of the best performing method. Following your suggestion, we have performed a two-sample t-test between the highest mean and the others, which changes only two cells in Table 1 and one in Table 2. The bolding will be changed to reflect this and will be explained in the captions.
>
> - All of the estimators are unbiased and only differ in their variance. Furthermore, if we consider model vs. method hyperparameters, none of the gradient estimators have any method hyperparameters to tune (such as the temperature parameter in Gumbel Softmax that requires careful tuning). Regarding the model hyperparameters like learning rate and network structure, we fix this for all methods and follow the experimental setup in previous work on discrete gradient estimators.
>
> - Thank you for bringing the attention to other recent work on Rao-Blackwellizing discrete gradient estimators, which we will add to the related work.

---

### Official Review · Reviewer_m9re · 2021-07-19

**Rating:** 7
**Confidence:** 4

**Summary:**

The paper proposed a novel gradient estimator for categorical stochastic hidden variables, named Categorical-Antithetic REINFORCE Multi-Sample (CARMS), which is a categorical extension of ARMS (Dimitriev and Zhou, 2021). CARMS reduces the variance of LOORF (Kool et at., 2019) by the means of importance weighted sampling. Using copula, where Dirichlet copula was employed in the experiments, CARMS generates the antithetic categorical samples to reduce variance. The paper compares two versions of the estimator, using inverse CDF sampling and Gumbel max sampling, against UNORD, LOORF and ARSM for categorical VAE and structured prediction tasks on Dynamic MNIST, Fashion MNIST and Omniglot. They find slightly better results compared to other estimators on both tasks.

**Limitations And Societal Impact:**

The authors adequately addressed the limitations and potential negative societal impact of their work.

**Main Review:**

### Originality:
There are a variety of approaches proposed in the past few years for unbiased gradient estimators over binary stochastic hidden variables. The paper introduces a natural extension for one of previously proposed estimators, ARMS, to categorical variables. Background and related works are adequately cited.
### Quality:
The authors did a great job in derivation of the estimators, design and implementation of the experiments. By closely checking the details, the methods are well-grounded.
### Clarity:
The paper, especially the method section, is clearly written and well explained.
### Significance:
The method proposed can be directly applied to interpretable models and modeling natural systems with discrete variables, e.g. sparse computing.

## Typos and potential places to improve:
* Ln 95, which both -> both of which
* Ln 148, adding a definition of Z^{-d}
* Eq 7, for l_i^o, the upper limit for the summation should be i-1, instead of j-1
* Figure 1, how many samples are used for estimating the correlation matrix? Also, how to interpret the numbers on two-axis of the matrices, as z[i] and z[j] should be categorical variables?
* Ln 198, “For CARMS-G, we clip the empirical ratio values to avoid numerical instabilities.” It is a bit unclear to me: 1. What is “the empirical ratio values”? Is it the sampled value for the ratio tensor R? 2. How did you clip it? E.g. clipping range, etc. I didn’t find the relevant information in either the main text or the appendix.
* In Figure 2 and 3, there are labels of “CARMS” and “ECARMS”, while in other places of the paper, the proposed methods are mentioned as CARMS-I and CARMS-G. Which CARMS-? Is “CARMS” referring to, and what does “ECARMS” stand for?

## Questions:
* In Ln 199 of the Experiment results section, it mentioned that the Dirichlet copula is used for the experiments? I am wondering whether there are results by trying Gaussian copula. To my limited knowledge, I find the Dirichlet copula introduces a limited dependence structure, while the Gaussian copula seems more flexible. Is there any insight into which copula to choose?
* In Ln 234 of the Categorical VAE section, it was mentioned that “an individual run took 5-8 hours” and it was on CPUs. This is surprisingly fast. Based on the description follows, it seems this holds for all estimators in Table 2 other than UNORD with C=10. Does this speed number also hold for ARSM? Also, in Line 245, it mentioned the number of evaluations is limited to 2C for ARSM, but in the original ARSM paper, it requires C(C-1)/2 evaluations for the swapping matrix. How does it (using 2C evaluations rather than the C(C-1)/2) impact the performance of ARSM as a baseline?
* The experiments show that CARMS-I/G “result in slightly higher log likelihood than other methods”. Is there any insight or result for the case when the number of categories is growing larger? It is common to have a model with C >> 10. I am curious, when C grows larger, shall I expect to see a larger gap between LOORF and CARMS or smaller?
* Eq 5 introduces a very general form of couplings of categorical variables p(z, z’). In Figure 1, the author compares correlation matrices of 5 different couplings. It seems inverse CDF sampling and Gumbel max sampling favor quite different sets of joint samples. Is there any insight on what kind of correlation structure is better?
* Based on the results, it seems LOORF is almost as good as the estimators using coupled samples. Does it mean that using the independent samples is good enough? Or putting the question in another way, in what situation, shall we expect a big improvement by using coupled samples?



**Time Spent Reviewing:**

6

---

> ### Author Response · Authors · 2021-08-10
> **Response to reviewer m9re**
>
> Thank you for your thoughtful and detailed review. We will fix all the typos, and we respond to each point below:
>
> - Regarding Figure 1, we used 1,000 samples to estimate it, and will mention this in the paper. We let $z_i$ or $z'_j$ be the $i^{th}$ or $j^{th}$ element of the onehot encoding of the categorical variables $z$ and $z'$, respectively. Marginally, both $z_i$ and $z'_j$ are Bernoulli variables, with two possible values 0 and 1, in which case we can calculate the correlation between any pair $(z_i, z'_j)$.
>
> - For CARMS to be unbiased we need a good estimate of the ratio $p(z_i)p(z_j) / p(z_i, z_j)$, for which we need to be able to estimate the bivariate probability mass function (PMF) in the denominator well. If we draw N samples to do this empirically, there is a small probability that some pair (i, j) does not occur at all, so we clip the maximum value of the ratio to 10 to avoid infinities. We will mention the exact value in the paper.
>
> - We apologize for the ECARMS labeling error, which stood for Empirical CARMS, the Gumbel version. In reading the results section, you may substitute CARMS and ECARMS with CARMS-I and CARMS-G, respectively, and this will be fixed in the final version.
>
> - Similarly to the ARMS paper, we did not find much of a difference in the experiments. We opted to use the Dirichlet copula because its analytical calculation resulted in twice as fast experimental running time. Although the Gaussian copula's correlation can take any value, we are interested in copulas that produce maximal negative dependence. Therefore, flexibility is not the main goal, but large negative correlation between each variable, to reduce duplicate samples.
>
> - Although the experiments were run on a CPU and finished quickly, some operations were automatically parallelized across its 56 cores, and we ran for one method all 3 datasets and both linear and nonlinear networks, simultaneously. If we force the run on a single core, a single run took 14-18 hours, which will be stated in the paper.
>
> - For ARSM, we noticed that in practice, using one sample with its corresponding $C(C-1)/2$ swaps resulted, on average, in $2C$ unique function evaluations per step. ARSM is upper bounded by $C(C-1)/2$ but the number varies empirically, and was on average 2C throughout a run for one sample. Therefore, we compared 1 sample ARMS to the other methods that used exactly C samples per step. The lower performance is most likely because those $C(C-1)/2$ swaps result in correlated and redundant samples, whereas our approach strives to maximize the number of different samples. We will make this clearer in the paper.
>
> - Regarding the performance gap when C grows larger: this is a good point for which we can offer some intuition. The gap between LOORF and CARMS should depend more on the entropy of the categorical variable than the number of categories C. It is true that if we assume a more uniform distribution over probabilities, e.g. close to 1/C, then the probability of a duplicate sample pair is $C$ times $1/C^2$ and grows ever more unlikely as C gets larger. However, even for a large number of categories, if one or a few categories have a large probability, this will put significant mass on those pairs being duplicates. We will add a paragraph describing this.
>
> - It is possible that having larger negative correlation for common categories would result in lower variance, but this will also depend on the magnitude of the function $f(z)$. We use the same negative correlation between each pair as it greatly simplifies sampling and debiasing, but future work may include a more flexible function-dependent structure or attempt to find an approach that produces strictly stronger negative correlations.
>
> - The improvement of our method using coupled samples should, as stated above, be larger when one or a few of the categories are large. In this case, the probability of a duplicate sample is large and can benefit from the negative correlations.

---

> > ### Comment · Reviewer_m9re · 2021-08-24
> > **Thanks for the detailed reply!**
> >
> > Thanks for the detailed reply.
> > Just one more question:
> > With multi-sample estimator, it is possibly reasonable to use multi-sample bound, e.g. in VIMCO (Mnih and Rezende [2016]) and ARMS (Dimitriev and Zhou [2021]). Is there any particular advantage of using multiple samples but training with ELBO? If using multi-sample objectives, and comparing against VIMCO or other multi-sample baselines, how would you expect CARMS performs?

---

> > > ### Author Response · Authors · 2021-08-24
> > > **Good question.**
> > >
> > > This is a good question that has been tackled in other work. According to Rainforth et al. (2018), it is not necessarily better to use the IWAE bound when having access to multiple samples. They show that although being tighter, the multi sample bound decreases the signal to noise ratio, which can be detrimental to learning and can outweigh the benefits.
> > > We have left the adaptation of CARMS to multi sample bounds for future work, along with a more general framework for antithetic sampling for multi sample bounds, which is beyond the scope of this paper.
> > >
> > > Rainforth et al. - Tighter Variational Bounds are Not Necessarily Better (ICML 2018).

---

### Decision · Program_Chairs · 2021-09-27

**Decision:**

Accept (Poster)

**Comment:**

The submission proposes a low-variance gradient estimator for discrete random variables, extending recent work on binary to categorical. Whilst the reviewers raised concerns around the marginal improvement over existing methods, all agreed that the submission is clearly written and is a theoretically grounded, novel extension of ARMS. Overall, this is a good contribution to NeurIPS and should be accepted.